DATA RELEASE

# AortaSeg-60: an open real-world CT-angiography dataset of the aorta with automated segmentation masks and pathological variability

Dania El Rahal[1], David C. Rotzinger[1] and Guillaume Fahrni[1,*]

1 Department of Diagnostic and Interventional Radiology, Lausanne University Hospital and University of Lausanne, Rue du Bugnon 46, CH-1011, Lausanne, Switzerland

## ABSTRACT

We present AortaSeg-60, an open dataset of 60 real-world thoraco-abdominal CT-angiography scans of the aorta encompassing normal anatomy and pathological variations, designed for AI research, benchmarking, and educational purposes. The dataset is organized into six balanced categories: young normal, elderly normal, aortic aneurysms, aortic dissections, venous acquisition, and non-contrast acquisition, capturing realistic anatomical and pathological diversity.

All scans are provided in NIfTI format with fully automated aortic segmentation masks generated using TotalSegmentator, without manual correction, enabling evaluation of typical algorithmic errors and testing of refinement strategies. Two radiologists performed a technical validation to ensure dataset curation and correct category assignment.

AortaSeg-60 is publicly available on Zenodo under a CC0 license. By providing paired imaging and automated labels, the dataset facilitates reproducible research, algorithm development, and method comparison for vascular segmentation, while noting limitations of sample size, single-centre acquisition, and reliance on automated annotations.

**Subjects** Imaging, Biomedical Science, Machine learning

**Submitted:** 03 February 2026

* Corresponding author. Email: guillaume.fahrni@chuv.ch

Preprint submitted at https://doi.org/10.20944/preprints202605.1002.v1

## INTRODUCTION

The development of robust artificial intelligence (AI) systems in medical imaging is strongly dependent on the availability of large, well-annotated, and openly accessible datasets. However, despite rapid progress in imaging technologies and machine learning methods, a persistent scarcity of openly shared medical imaging datasets persists [1]. This limitation restricts reproducibility, slows methodological progress, and creates barriers for independent validation of published algorithms.

Among the different forms of annotation, segmentation datasets play a central role in the advancement of AI-driven image analysis. Pixel- or voxel-level labels enable not only supervised model training but also rigorous validation and standardized testing across institutions and research groups. Such datasets are essential for the development of clinically relevant tools, for quantitative anatomical analysis, and for fair comparison between competing algorithmic approaches [2].

To mitigate the lack of publicly available data, several initiatives have turned toward the generation of synthetic datasets [3]. While synthetic imaging can provide large volumes of labelled data emulating a range of acquisition variables, anatomical or pathological features at a relatively low cost, it may also introduce distributional biases, unrealistic anatomical variations, or imaging artifacts that do not accurately reflect clinical conditions. Consequently, synthetic approaches are best considered complementary rather than substitutive to real clinical data.

Real-world clinical datasets remain critical for capturing the heterogeneity inherent to medical imaging, including variability in imaging systems, protocols, contrast timing, demographic factors, and pathological presentations. For example, the aorta, the largest artery in the body extending from the thorax to the pelvis, normally has a tubular shape, but its morphology can vary substantially in the presence of pathologies such as aneurysm, dissection, or age-related changes. Models trained and validated only on normal aortas are therefore likely to underperform when confronted with this real-world variability. Clinical validation of algorithms on heterogeneous datasets is thus a necessary step toward responsible translation into healthcare settings [4].

Widely used automated segmentation frameworks, such as TotalSegmentator, are designed to generalize across imaging conditions and variability, having been trained on diverse multi-institutional datasets encompassing different scanners and protocols [5]. In this context, independent datasets that focus on specific organs and include various pathological conditions can improve the evaluation and refinement of such tools.

Despite the clinical importance of vascular imaging, publicly available datasets dedicated to the aorta remain scarce compared to other organs. Existing resources are often highly specialized or limited in scope. For example, the AVT dataset focuses primarily on the geometric tree structure across multiple centres, while ImageTBAD and ImageTAAD are restricted to specific emergency pathologies, namely Type-B and Type-A dissections [6–8]. Furthermore, large-scale multi-organ frameworks like TotalSegmentator provide broad anatomical coverage but lack granular focus on the diversity of aortic acquisition protocols. Currently, there is a distinct lack of open-access datasets that bridge the gap between normal and pathological anatomy while simultaneously accounting for varied imaging conditions, such as different contrast injection phases (arterial, venous, and non-contrast) and ECG-gating.

In this context, the objective of the present dataset is to provide a collection of open ECG-gated computed tomography angiography (CTA) scans of the aorta, acquired under heterogeneous clinical conditions, including variability in patient age, pathological presentation, and contrast injection timing, thereby reflecting real-world clinical diversity. The dataset also provides automated segmentation masks generated with TotalSegmentator, allowing users to assess labelling accuracy, explore refinement strategies, and compare performance across different segmentation algorithms.

## DATA ACQUISITION

This retrospective dataset was collected at a single tertiary academic hospital and consists of 60 CTA examinations acquired between 2014 and 2024. The whole dataset creation pipeline is summarised in Figure 1.

Cases were manually curated from the institutional Picture Archiving and Communication System (PACS) using a two-step selection process that combined

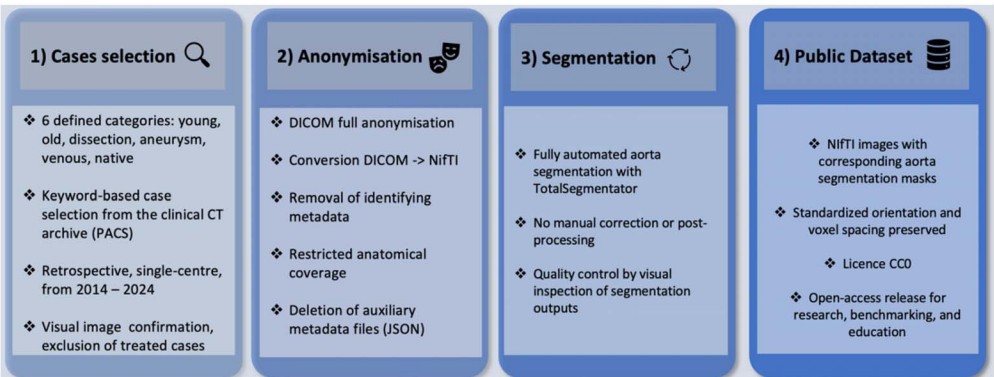

**Figure 1.** Dataset creation pipeline. CT cases were selected from a single-centre archive (2014–2024) and categorized (young, old, dissection, aneurysm, venous, non-contrast). Data were fully anonymised, converted to NIfTI, and segmentations of core organs were generated. The resulting dataset, including images and masks, is publicly available.

keyword-based screening of radiology reports (for example "aortic aneurysm", "aortic dissection", and "aorta") followed by direct visual confirmation on the images to verify either normal any or the presence of aortic pathology.

The dataset was intentionally organized into six balanced categories, each comprising ten examinations: (1) young patients (<30y, arterial acquisition) with normal aorta (2) elderly patients (>70y, arterial acquisition) with normal aorta or age-related changes; (3) aortic aneurysms (arterial acquisition); (4) untreated Stanford type A or type B dissections (arterial acquisition); (5) venous acquisition without significant aortic pathology; and (6) non-contrast acquisition without significant aortic pathology. The first two categories (1–2) were selected to represent normal aortic anatomy across a spectrum of age-related and atherosclerotic changes. The aneurysm and dissection groups (3–4) provide challenging cases where automated algorithms may struggle with severely distorted morphology and complex luminal alterations. Finally, the venous and non-contrast categories (5–6) were included to assess algorithmic robustness in scenarios of suboptimal or absence of aortic opacification. Example of patient images of the six categories are provided in Figure 2. This balanced sampling strategy was specifically chosen to ensure equal representation of diverse clinical scenarios, providing a controlled environment for methodological testing. Consequently, the dataset's composition does not reflect the natural prevalence or epidemiological distribution of aortic pathologies in a clinical population.

All included subjects were adults. Imaging was performed using CT systems from a single vendor (GE Healthcare) in order to limit hardware-related variability while still preserving heterogeneity in acquisition protocols patient characteristics. All examinations, including venous and non-contrast series, were ECG-gated. Typical acquisition parameters included a 128 to 256-slice configuration, 0.60 to 1.25 mm slice thickness, a 512 × 512 reconstruction matrix, tube voltage ranging from 80 to 120 kVp, and automatic tube current modulation. Standard injection timing was 40 s. for the arterial phase and 90 s. for the venous phase. The field of view was typically centered on the thoraco-abdominal and images were reconstructed using a standard soft-tissue kernel. These parameters reflect routine clinical practice in our institution and in broader cardio-vascular imaging standards. Cases description and assoiated acquisition parameters are summarised in Table 1.

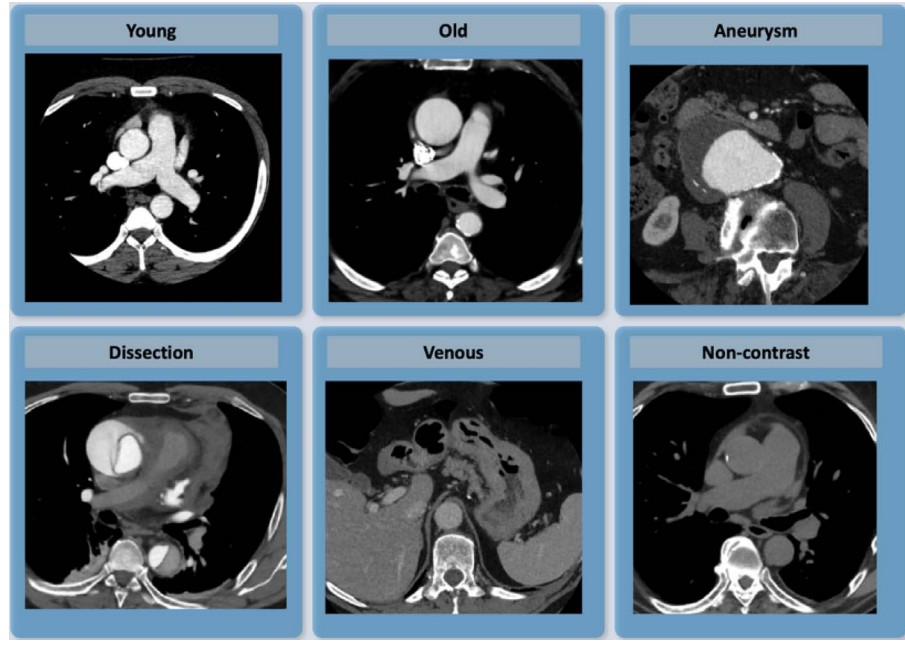

**Figure 2.** Representative axial CT slices (mediastinal window) for each of the six categories: young normal, elderly normal, aortic aneurysms, aortic dissections, normal venous acquisition, and normal non-contrast acquisition.

**Table 1.** Summary of the AortaSeg-60 dataset by category, including key CT acquisition parameters.

| Category | N | Matrix size (voxels) | Voxel spacing (mm) | Slice thickness (mm) | In-plane FOV (mm) |
|---|---|---|---|---|---|
| Aneurysm | 10 | 512 × 512 × 1040–1161 | 0.50 ± 0.10 (0.42–0.77) | 0.60 ± 0.00 (0.60–0.60) | 256.8 ± 52.5 (216.0–396.0) |
| Dissection | 10 | 512 × 512 × 528–1278 | 0.58 ± 0.15 (0.43–0.82) | 0.73 ± 0.27 (0.60–1.25) | 299.1 ± 77.6 (219.0–421.0) |
| Non-contrast | 10 | 512 × 512 × 1007–1198 | 0.66 ± 0.16 (0.44–0.81) | 0.60 ± 0.00 (0.60–0.60) | 339.0 ± 79.6 (227.0–414.0) |
| Old | 10 | 512 × 512 × 510–1148 | 0.60 ± 0.12 (0.49–0.79) | 0.73 ± 0.27 (0.60–1.25) | 305.4 ± 63.8 (249.0–402.0) |
| Venous | 10 | 512 × 512 × 1015–1148 | 0.55 ± 0.08 (0.43–0.74) | 0.60 ± 0.00 (0.60–0.60) | 280.0 ± 43.2 (220.0–379.0) |
| Young | 10 | 512 × 512 × 538–1153 | 0.50 ± 0.08 (0.39–0.67) | 0.73 ± 0.27 (0.60–1.25) | 255.8 ± 40.1 (201.0–345.0) |

## ETHICAL COMPLIANCE AND ANONYMIZATION

The Legal Affairs Unit of CHUV has confirmed that the AORTASEG study (project Hors-LHR-BPR1247) falls outside the scope of the Swiss research legislation (Federal Act on Research involving Human Beings, Human Research Act, HRA, SR 810.30) and therefore does not require ethics committee authorization, in accordance with Swiss legislation and institutional guidelines.

Prior to public dissemination, all imaging data underwent a strict anonymization procedure. Identifying DICOM metadata fields were removed or replaced, and only fully de-identified image volumes were exported. The anatomical coverage was restricted to the thoracic and abdominal regions, thereby excluding facial structures and further reducing any potential re-identification risk. To further strengthen full anonymisation, reduce dataset size and provide research-ready images, conversion from DICOM to NifTI format was performed for all cases, and residual JSON medata files from the NifTI images were deleted. The final dataset complies with institutional data-protection standards and commonly accepted open-data sharing practices in medical imaging research.

## SEGMENTATION PIPELINE

Automated aortic segmentations were generated using the open-source TotalSegmentator framework, version 2.12.0. The dedicated aorta task (task identifier 52) was executed through a Python-based pipeline (python version 3.10.19) on a local workstation equipped with a modern graphics processing unit. No image pre-processing, resampling, or intensity normalization steps were applied prior to segmentation in order to preserve the original clinical characteristics of the scans.

All segmentation masks were produced fully automatically and were not manually corrected or edited. This design choice was intentional, as the objective of the dataset is not to provide expert ground-truth contours but rather to reflect realistic automated labeling conditions. The resulting masks therefore allow users to evaluate typical automated segmentation errors, test post-processing or refinement strategies, and benchmark alternative algorithms under conditions that are closer to real-world deployment scenarios. Given the open-source nature of TotalSegmentator, providing raw data allows for future updates as the algorithm evolves. Recent iterations have already demonstrated significant improvements in cardiovascular segmentation accuracy [9], making this dataset a valuable tool for tracking AI progress.

## TECHNICAL VALIDATION

A basic technical validation of the dataset was performed independently by two board-certified radiologists with experience in cardiovascular imaging. The objective was to ensure dataset consistency and usability. The validation consisted of a systematic visual inspection of all cases.

For each examination, the reviewers confirmed the visible presence of the aortic segmentation mask and verified that the segmentation covered the expected anatomical region of the thoracic and abdominal aorta. In addition, the clinical category assigned to each case was checked against the image content and the associated radiological report to ensure correct labeling. Particular attention was given to avoiding cross-category inconsistencies, such as the co-existence of aneurysm or dissection in cases labeled in another category, or incorrect attribution of contrast phase categories.

## REUSE POTENTIAL AND APPLICATIONS

The dataset is designed to support research and educational applications in medical image analysis. Its structure and accompanying automated segmentation masks make it suitable for supervised or semi-supervised AI model training and finetuning, particularly for tasks related to vascular detection and segmentation. The presence of heterogeneous acquisition conditions and multiple pathological categories also enables benchmarking of segmentation algorithms under varying clinical scenarios.

Beyond direct algorithm development, this dataset may also serve educational purposes in radiology and biomedical engineering training environments, where realistic clinical images and automated labels can help illustrate both normal anatomy and common vascular pathologies or segmentations errors. In addition, the dataset is appropriate for testing preprocessing pipelines, data loading frameworks, and method comparison studies that require consistent paired image–label data without the necessity of large-scale expert annotations.



## LIMITATIONS

Several limitations should be considered when using this dataset. First, all segmentation masks were generated automatically and were not manually corrected by design, which implies the presence of typical algorithmic errors and variability in contour precision. Second, the overall sample size remains limited to 60 cases, which may restrict statistical power for certain machine learning or epidemiological analyses. Furthermore, although efforts were made to include heterogeneous acquisition contexts, the dataset originates from a single institution and a single CT system vendor, which may limit generalizability across hardware ecosystems. Finally, the dataset is intended for research, benchmarking, and educational purposes and is not designed or validated for direct clinical decision making or diagnostic use.

## CONCLUSIONS

This work presents an openly accessible and curated dataset of 60 real-world clinical thoraco-abdominal CT-angiography examinations focused on the aorta, accompanied by automated segmentation masks. The dataset aims to facilitate reproducible research in medical image analysis by providing paired imaging and labeling data that reflect realistic clinical variability. By enabling benchmarking, algorithm comparison, and methodological experimentation, it is intended to encourage community reuse, extension, and transparent evaluation practices in AI-driven radiological research.

## DATA AVAILABILITY

The dataset is publicly hosted on Zenodo [10], with a total compressed size of approximately 16 GB. The data are organized into category-based folders corresponding to the six clinical groups, with ten cases per category. For each case, two primary files are provided: the original CT volume and the corresponding automated aortic segmentation mask. Both are distributed in NIfTI format to ensure broad compatibility with common medical image analysis software.

Each category is distributed as a compressed archive, and a consistent file-naming convention is applied across all cases to facilitate automated processing. The dataset is released under a CC0 (Creative Commons Zero) 1.0 Universal license and follows a versioned release scheme to ensure long-term citability and reproducibility.

## DECLARATIONS

### Ethics approval and consent to participate

The authors declare that ethical approval was not required for this type of research.

### Competing interests

The authors declare that they have no competing interests related to this work.

### Authors' contributions

DER: Conceptualization, dataset design, data curation, case selection, technical validation, manuscript writing and editing. GF: Conceptualization, project supervision, manuscript review and revision. DCR: Clinical validation, supervision, manuscript review and revision.

### Funding

Not applicable.

## Acknowledgements

The authors acknowledge the use of artificial intelligence-based writing assistance tools (ChatGPT, OpenAI) for formatting support and language refinement during manuscript preparation.

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
