## [Reviewer Report]

Indicate in the comments box below whether you are happy with the changes made or if the manuscript is unacceptable.Comments on revised manuscriptThank you for the authors’ response and revisions. I am supportive of this revised version. I encourage the authors to further provide a high-quality dataset with manually reviewed/corrected annotations through the “Update Description” process in the future.Indicate in the comments box below whether you are happy with the changes made or if the manuscript is unacceptable.Comments on revised manuscriptThank you for the authors’ response and revisions. I am supportive of this revised version. I encourage the authors to further provide a high-quality dataset with manually reviewed/corrected annotations through the “Update Description” process in the future.

---

## [Editor Report]

Editor’s AssessmentThe manuscript is ready for formal acceptance.Editor’s AssessmentThe manuscript is ready for formal acceptance.

---

## [Reviewer Report]

Reviewer name and names of any other individual's who aided in reviewer Maurice PradellaDo you understand and agree to our policy of having open and named reviews, and having your review included with the published papers. (If no, please inform the editor that you cannot review this manuscript.)YesIs the language of sufficient quality?YesPlease add additional comments on language quality to clarify if needed
minor adjustments necessaryAre all data available and do they match the descriptions in the paper? YesAdditional CommentsAre the data and metadata consistent with relevant minimum information or reporting standards? See GigaDB checklists for examples <a href="http://gigadb.org/site/guide" target="_blank">http://gigadb.org/site/guide</a>YesAdditional CommentsIs the data acquisition clear, complete and methodologically sound?NoAdditional Commentsconsider rephrasing the categories to be focussed on based on scan parameters: 1) ECG-gated CTA in normal aorta in young patients, 2) elderly patients and 3) acute aortic syndrome/dissection as well as 4) CT chest (abdomen) in venous phase and non-contrast
"radiologists confirmed the presence of segmentation". you state that there was no contour adjustment, so radiologists just checked whether contours are present seems like they are overqualified for the job. Consider rephrasing. (I personally don't understand why you needed two radiologists but you don't have to reply to this) I suggest not to write "venous-phase aortic CTA" since CTA refers to arterial imaging, please rephrase lastly, please explain why you decided not to provide adjusted contours as gold standard.Is there sufficient detail in the methods and data-processing steps to allow reproduction?YesAdditional Commentsyou can add that since the totalsegmentator is open source, segmentations be performed by an updated version in the futureIs there sufficient data validation and statistical analyses of data quality? YesAdditional Commentsplease consider adding: Hinck D, Segeroth M, Miazza J, Berdajs D, Bremerich J, Wasserthal J, Pradella M. Automatic Segmentation of Cardiovascular Structures on Chest CT Data Sets: An Update of the TotalSegmentator. Eur J Radiol. 2025 Apr;185:112006. doi: 10.1016/j.ejrad.2025.112006. Epub 2025 Feb 15. PMID: 39983596. as a reference since it significantly improved the aortic segmentations of the TotalSegmentatorIs the validation suitable for this type of data?YesAdditional Commentsnot necessary, TotalSegmentator provides stable segmentations of the aorta in generalIs there sufficient information for others to reuse this dataset or integrate it with other data?NoAdditional Commentsconsider adding DICOM versions of casesAny Additional Overall Comments to the AuthorThank you very much for your paper. I think, your data set contributes significantly to the field aortic imaging.RecommendationMinor Revision

---

## [Reviewer Report]

Reviewer name and names of any other individual's who aided in reviewer Zekun JiangDo you understand and agree to our policy of having open and named reviews, and having your review included with the published papers. (If no, please inform the editor that you cannot review this manuscript.)YesIs the language of sufficient quality?YesPlease add additional comments on language quality to clarify if needed
Are all data available and do they match the descriptions in the paper? YesAdditional CommentsAre the data and metadata consistent with relevant minimum information or reporting standards? See GigaDB checklists for examples <a href="http://gigadb.org/site/guide" target="_blank">http://gigadb.org/site/guide</a>YesAdditional CommentsIs the data acquisition clear, complete and methodologically sound?NoAdditional CommentsThe 60 cases provided by the authors were manually designed and divided into six major subgroups, with 10 cases in each group. In fact, this setup does not reflect real-world distributions, so the title itself is open to question. This is essentially an artificially designed dataset rather than one that truly represents real-world patterns.Is there sufficient detail in the methods and data-processing steps to allow reproduction?YesAdditional CommentsIs there sufficient data validation and statistical analyses of data quality? YesAdditional CommentsIs the validation suitable for this type of data?NoAdditional CommentsLack of high-quality segmentation annotations.Is there sufficient information for others to reuse this dataset or integrate it with other data?YesAdditional CommentsAny Additional Overall Comments to the AuthorThis manuscript presents AortaSeg-60, an open dataset containing 60 thoraco-abdominal aortic CT/CTA scans, covering six representative clinical scenarios and providing aortic segmentation masks automatically generated by TotalSegmentator. The authors clearly describe the data source, categorization strategy, anonymization process, segmentation pipeline, public sharing platform, and potential research applications. As an open real-world data resource, this work has certain value, particularly for algorithm testing, post-processing improvement, weak-label research, and educational purposes. The overall logic of the manuscript is clear, and the importance of data sharing is evident. However, I believe there are still several issues in the current manuscript that require further clarification and strengthening: 1. The biggest limitation of this dataset is the lack of manual correction, which greatly affects the overall data quality. As a result, the dataset cannot be used directly by other researchers as a reliable reference. In other words, all segmentation masks were generated automatically by TotalSegmentator and were not manually revised, making this dataset closer to a weak-label/automated-label dataset than to a carefully curated ground-truth segmentation benchmark. 2. Although the category design is reasonable, it does not represent the true clinical distribution and should be stated more explicitly. The six categories with 10 cases each reflect an intentionally balanced sampling strategy, which is suitable for methodological testing. However, this design does not reflect real-world disease spectra or clinical prevalence. The authors are encouraged to state more clearly in the Methods or Discussion that: this was a balanced design intended to cover diverse clinical scenarios; the dataset is not suitable for epidemiological or representative population analysis; its value lies more in controlled heterogeneity testing rather than real-world distribution sampling. 3. There are inconsistencies between the text and the figures that should be corrected. The most obvious issue concerns the license statement. The main text states that the dataset is released on Zenodo under a CC0 license, whereas the fourth box in Figure 1 states “License CC-BY 4.0.” 4. The comparison with existing publicly available aortic/vascular datasets is still insufficient.RecommendationMajor Revision